# RNA-Pol II Transcription Elongation Factor FpRtfA Is Required for Virulence and Secondary Metabolism in *Fusarium pseudograminearum*

**Yuxing Wu, Yajiao Wang, Sen Han, Qiusheng Li and Lingxiao Kong ***

Institute of Plant Protection, Hebei Academy of Agricultural and Forestry Sciences, Integrated Pest Management Center of Hebei Province, Key Laboratory of IPM on Crops in Northern Region of North China, Ministry of Agriculture, Baoding 071000, China; wyx1209@haafs.org (Y.W.); yajiaowang515@163.com (Y.W.); hansenmayun@sina.com (S.H.); alidd@163.com (Q.L.)
* Correspondence: konglingxiao163@163.com

**Abstract:** The soil-borne pathogen *Fusarium pseudograminearum* is capable of causing a highly destructive crown disease in wheat. The purpose of this study was to characterize the biological functions, such as of virulence and secondary metabolites (SMs), of a putative RNA-Pol II transcription elongation factor, FpRtfA, in *F. pseudograminearum*. The current study revealed that the deletion of *FpRtfA* reduced radial growth compared to wild type in *F. pseudograminearum*. In addition, *FpRtfA* was found important to conidiation and response to metal ions and oxidative stress. More importantly, the virulence to the wheat stem base or head was decreased when *FpRtfA* was deleted. Using genome-wide gene expression profiling, *FpRtfA* was found to regulate several processes related to the above phenotype, such as the carbohydrate metabolic process, and the response to oxidative stress and oxidoreductase activity, especially for SMs. Further, we proved that FpRtfA exerts its regulatory effect on the virulence of pathogens by modulating the expression of the PKS gene, but not the generation of DON. In conclusion, FpRtfA has important roles in growth, asexual development, and the response to metal ions and oxidative stress. More importantly, FpRtfA is required for SMs and full virulence in *F. pseudograminearum*.

**Keywords:** *Triticum aestivum*; deletion mutant; conidiation; secondary metabolite; *Fusarium pseudograminearum*

## 1. Introduction

*Fusarium* crown rot (FCR) is a highly destructive disease in arid and semiarid wheat regions of the world. Under natural inoculum levels, losses of FCR can be up to 10%~35% of the yield [1,2]. In the past decade, this global disease has become more common in the Huang-Huai areas of wheat production of China, including the Henan, Shandong, and Hebei provinces [3,4]. The typical symptom of FCR in wheat is browning of the leaf sheaths and subcrown internodes following colonization of the pathogen in coleoptiles in the field. Severely diseased wheat plants can develop white heads with shriveled grains due to necrosis at the base of the stem, causing significant yield losses [5]. *Fusarium pseudograminearum* is the primary FCR pathogen in several warm and dry wheat regions of the world, including the Australian wheat belt, the Pacific Northwest of the USA, and the Huang-Huai wheat-growing region of China [6,7].

Plant pathogenic fungi usually secrete diversified secondary metabolites (SMs) to establish colonization and develop pathogenicity at different stages of infection of the plant hosts. These SMs include mycotoxins and secretory proteinaceous toxins. For plant pathogenic fungi with different lifestyle or range of genera, SMs also play different roles. Host specificity of fungi is conferred through the production of host-specific toxins (HST) [8]. The difference is that in *Fusarium* spp., some SMs also play important roles in development

and pathogenicity. For example, *F. graminearum*, which causes Fusarium head blight (FHB) in small grain cereals, can produce various trichothecene mycotoxins, including deoxynivalenol (DON), nivalenol (NIV), and polyketide zearalenone, during infection of plant tissues [9]. The trichothecene mycotoxin DON is a potent protein synthesis inhibitor in eukaryotic organisms, and also acts as a significant virulence factor in infection [10,11]. Similar to *F. graminearum*, *F. pseudograminearum* produces a number of SMs that can act as pathogen virulence factors. Reduced pathogen virulence was observed in the tri5 mutant, suggesting that DON plays a role in the virulence of *F. pseudograminearum* [12–14]. In addition to DON, the *F. pseudograminearum* genome is predicted to encode at least 16 NRPSs and 14 PKSs that can participate in the synthesis of a variety of secondary metabolites in fungi. This implies that other SMs should affect the development and pathogenicity of fungi [12,15]. Analyses of function or regulation of SMs could elucidate common genetic links with the developmental or pathogenic lifecycle of *F. pseudograminearum*.

The regulated modes of fungal SM biosynthesis include signal transduction pathways, epigenetic modifications, global regulators, and specific regulators [16]. Global regulators, including in the response to carbon and nitrogen sources, ambient light, and pH, have been identified in several fungi. Thereinto, light affected the expression of fungal SMs through the velvet family protein complex VelB/VeA/LaeA. The velvet complex in SM biosynthesis has been characterized in various filamentous fungi. However, regulation mechanisms are needed further study. They are possible target key regulators of fungal SMs for developing new methodologies to reduce the negative effects of phytopathogens [11]. On that account, one of the VeA-dependent genetic elements, named RtfA, was identified in the model fungus *A. nidulans* utilizing a mutagenesis screening method. The RtfA was first characterized in *Saccharomyces cerevisiae* as an RNA-Pol II transcription elongation factor. This protein is important for TATA site selection by TATA box-binding protein. In addition, but not limited to this, it is involved in ubiquitination and dimethylation of histone H2B and H3, respectively. In filamentous fungi, RtfA has been found to regulate morphogenesis, development, and production of some SMs in *Aspergillus nidulans*, *A. fumigatus*, and *A. flavus* [17–21]. RtfA is considered a global regulator that affects AFB1, paxilline, aflatrem, aflavinines, leporins, aspirochlorine, and ditryptophenaline in *A. flavus*. More importantly, it regulates several processes, including tolerance to oxidative stress, cell wall composition and integrity, protease activity, and adhesion to surfaces. These functions could be important for the successful fungal infection of host tissue [22].

In considering the regulatory effects of RtfA on SMs in *Aspergillus* spp., it is necessary to understand the intricate roles of RtfA in the regulation of SMs as important virulence factors in the pathogenic fungi *F. pseudograminearum*. In addition, the mechanism of this regulation may need further research to further elucidate the development and pathogenicity in *F. pseudograminearum*.

In this study, we examined the effects of the RtfA orthologous gene *FpRtfA* on the growth, conidiation, virulence, and expression of SM genes in *F. pseudograminearum*. The deletion of *FpRtfA* caused a difference in conidiation. Moreover, attenuated virulence was detected in *FpRtfA* gene deletion mutants. *FpRtfA* was also found to regulate the expression of multiple metabolic pathway genes, particularly SMs. Though the generation of DON was independent of *FpRtfA*, another SM gene cluster is positively regulated by *FpRtfA* and is associated with pathogenesis. These results indicated that *FpRtfA* is involved in the development, virulence, and SMs of *F. pseudograminearum*.

## 2. Materials and Methods

### 2.1. Strains and Growth Conditions

The *Fusarium pseudograminearum* wild-type strains 2035 utilized in this study was preserved in the Laboratory of Fungi Diseases at the Institute of Plant Protection, Hebei Academy of Agricultural and Forestry Sciences, PRC. The wild-type and mutant strains were grown on a potato dextrose agar (PDA, per 1 L:200 g potato extract, 20 g dextrose, 15 g agar) medium at 25 °C unless otherwise specified.

The growth rates of different strains were expressed by the colony radius per day. Carboxymethylcellulose sodium (CMC) medium was used for the conidiation assay. Different strains were placed in a shake flask culture in 250 mL conical flask containing 100 mL CMC at 25 °C, 170 revolutions per minute (rpm) for four days. The concentration of conidia was determined using a hemocytometer [23]. Complete media (CM) were used to determine the stress responses of different strains. This essential medium was composed of 0.6% yeast extract *w/v*, 0.6% casein hydrolysate *w/v*, 0.2% peptone and 1% sucrose *w/v*, NaNO3 1.2% *w/v*, KCl 0.1% *w/v*, KH2PO4 0.3% *w/v*, and agar 1.5% *w/v*. The strains were incubated on CM media-mended NaCl (0.7 M), $H_2O_2$ (3 mM), Congo red (200 mg/L), or sodium dodecyl sulfate (SDS, 0.01%) for four days. TB$_3$ medium (yeast extract 0.3% *w/v*, 0.3% *w/v* casamino acids, sucrose 20% *w/v*, agar 1.5% *w/v*), supplemented with hygromycin B (250 µg/mL, Calbiochem, LaJolla, CA, USA) or geneticin (250 µg/mL, Sigma, St. Louis, MO, USA), was used to revive and select resistant transformants during gene deletion or complementarity.

## 2.2. Transformants of Deletion and Complementarity

*FpRtfA* was queried in the *F. pseudograminearum* genomic sequence (GenBank accession NC_031951.1) using the homologous alignment method. Conserved Domain Search Service (CD Search) of the National Center for Biotechnology Information (NCBI) was used to predict the conserved domains of *FpRtfA*. The ClustalW tool was used to align *FpRtfA* and its homologous proteins. The neighbor-joining method was used to construct the phylogenetic tree using the MEGA version 7.02 software package [24].

The deletion mutants were constructed via the homologous recombination method. The hygromycin phosphotransferase (*hph*) gene was used to replace open reading frames of target genes. Replacement fragments were constructed by joining upstream and downstream flanking fragments of the target genes and *hph* fragments via double-joint polymerase chain reaction (DJ-PCR) [25]. The upstream and downstream flanking fragments of the target genes were amplified with primer pairs 1F/2R and 3F/4R. The primer pair HYG-F/R was used to amplify the *hph* gene. The target replacement fragment was transformed into protoplasts of wild-type 2035 using the polyethylene glycol (PEG) approach. The preparation and transformation of protoplasts were performed as described [26]. After screening with the TB$_3$ medium supplemented with hygromycin B, the positive transformants were verified using the primer pairs 5F/6R, H850/852, 7F/H855R, and H856F/8R. In addition, Southern blot analyses confirmed the deletion mutants [27]. For complementation assays, the full length with the promoter region of target genes was amplified with the primer pair CF/CR. These fragments and XhoI-digested pFL2 were co-transformed into the yeast strain XK1-25 using yeast gap repair, as described in [28,29]. The target gene–pFL2 construct was transformed into respective deletion mutants using the PEG approach. The detection primer pair 5F/6R was used to identify successful complementarity. The primers used for deletion, complementarity, and gene expression are listed in Supplementary Table S1.

## 2.3. Assays for Virulence and Infection

Conidia of different strains were collected from CMC medium. The collected conidia were diluted to a suspension with a concentration of $10^5$ conidia/mL. The susceptible cultivar Shixin 828 was used for the infection test. For inoculation on the wheat stem base, the procedure, as described by Li et al. (2009), was carried out with the following modifications: Seeds of wheat were germinated on saturated filter paper in Petri dishes. Then, the germinated seeds were immersed in the conidial suspension for 1 min. The immersed seeds were sown in a pot with a diameter of 15 cm, containing a sterile soil mix. Each pot contained 20 seeds. Following planting, all the pots were incubated in a glasshouse with 60/80 (±10)% day/night relative humidity (RH) and 25/15 (±5) C day/night temperature. The severity of *Fusarium* crown rot was assessed 35 days post-inoculation (dpi) using a 0–5 scale [30].

In the wheat head infection test, a 20 μL aliquot of conidial suspension with $10^5$ conidia/mL was injected into a floret of a wheat head at early anthesis. Each strain was replicated 30 times. The severity of head blight was assessed on a scale of 0–4 [31].

The plants were designated a disease index value. The disease index (DI) was calculated as DI = [∑ (number of diseased plants in this scale × value of this scale) / (total number of plants investigated × highest value of scale)] × 100.

### 2.4. RNA-Seq and Bioinformatics Analysis

The colonies of different strains were incubated on PDA at 25 °C for 3 d. Mycelia samples were collected from the surface of the colonies. There were three biological replicates per treatment. A ribonucleic acid (RNA) extraction kit (Qiagen, Hilden, Germany) was used to extract the total RNA of the mycelium according to the manufacturer's instructions. The processes of library preparation and sequencing were performed by Novogene Co. Ltd. (Tianjin, China). TopHat 2.0.8 software with default parameters was used to map the clean reads in the reference genome of *F. pseudograminearum* CS3096 [32]. The gene expression was calculated by the number of reads per kilobase per million (RPKM) reads of the mapped read counts. The differences in gene expression between mutant and wild type were identified using HTSeq v0.9.1 software [33]. The differentially expressed genes (DEGs) were isolated with a false discovery rate (FDR) adjusted $p < 0.05$ via DESeq2 software [34]. The log2(fold change, FC) calculated from the RPKM value of the same gene being greater than 1.0 indicated FC between the mutant and wild type. The GOseq software package v1.24.0 was used to perform Gene Ontology (GO) annotation [35]. ClusterProfiler v3.8.1 software was used to analyze the Kyoto Encyclopedia of Genes and Genomes (KEGG) enrichment at $p < 0.05$.

Quantitative real-time PCR (qRT-PCR) was used to determine the transcript levels of SM genes [36]. For the expression of SM genes between wild type and mutants during infection, different strains were inoculated and cultivated as assays for virulence and infection. The tissues of stem bases were collected at 7 dpi. The total RNA was isolated from the mycelia or plant tissues using an RNA extraction kit (Qiagen, Hilden, Germany) according to the manufacturer's instructions. The assessment of integrity of RNA quality and concentration used agarose gel electrophoresis and an ultramicro spectrophotometer, respectively. First-strand cDNAs were synthesized from total RNA according to the manufacturer's instructions for the Fermentas 1st cDNA synthesis kit (Hanover, MD). All quantitative results were calculated using the $2^{-\triangle\triangle CT}$ method [37]. The expression quantities were normalized by the *β-tubulin* (TUB) gene [38]. The means and standard deviations of the data were collected from three biological replicates. The primers used are listed in Supplementary Table S1.

### 2.5. Determination of DON Production

Three mycelium agar plugs with a 6 mm diameter were inoculated into a 150 mL Erlenmeyer flask containing 30 mL trichothecene biosynthesis induction (TBI) medium [39]. The Erlenmeyer flasks were cultivated in a shaker at 180 rpm, 28 °C for 14 days. The fermentation broth was filtered with a 0.22 μm aqueous filter. The DON was detected via the ultra-performance liquid chromatography–tandem mass spectrometry (UPLC-MS/MS) method [40].

### 2.6. Data Statistics

Statistical analysis of all data used Fisher's least significant difference (LSD) in the Statistical Package for the Social Sciences (SPSS).

## 3. Results

### 3.1. Deletion and Complementarity of FpRtfA

The FpRtfA protein (accession number XP_009254491.1), containing 597 amino acids (aa), was identified as 36.45% homologous with *A. nidulans* RtfA (XP_662174.1). This

protein predicted the COG5296 super family transcription factor involved in TATA site selection and in elongation by RNA polymerase II via the CD Search of the NCBI. A plus-3 domain containing about 108 residues in length was located at aa 253–360. The *FpRtfA* gene sequence was interrupted by three introns at 186–321, 1044–1094, and 1230–1285 bp (Supplementary Figure S1A,B). Phylogenetic analysis showed that FpRtfA is a conservative RtfA homologue in filamentous fungi that is closely related to *Fusarium graminearum* (Figure 1A).

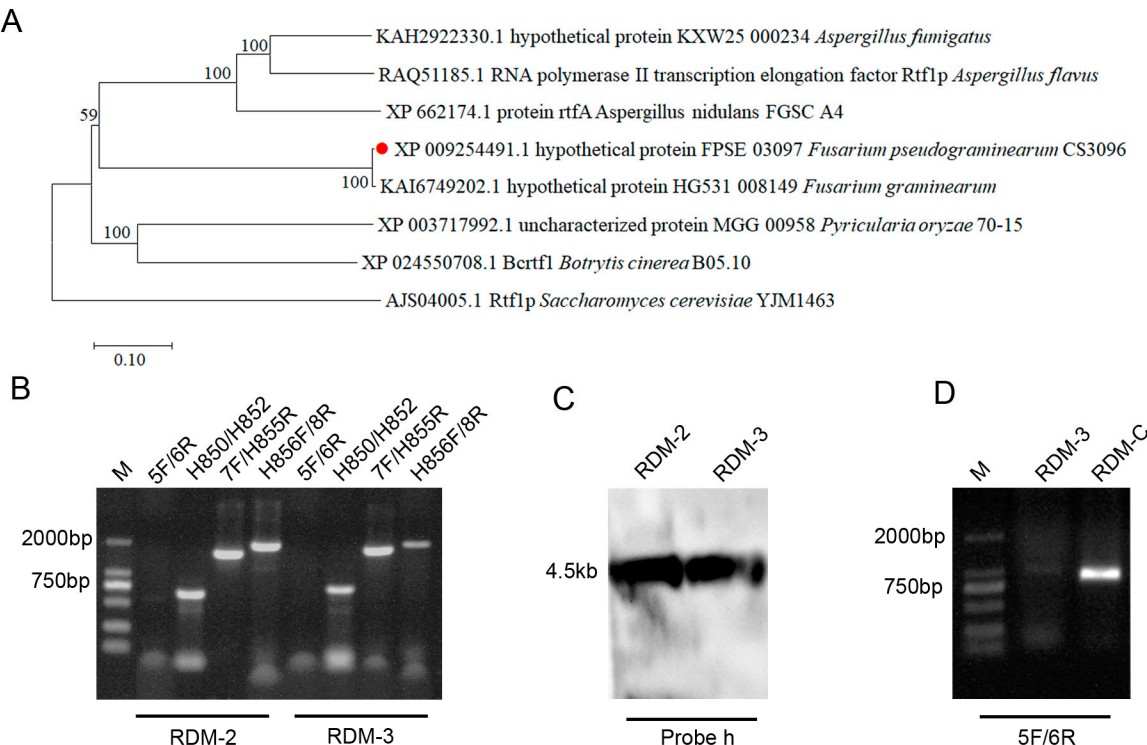

**Figure 1.** Phylogenetic analysis of FpRtfA and generation of FpRtfA gene deletion mutants. (**A**) Phylogenetic analysis of FpRtf of *F. pseudograminearum* (Marked with a red dot) and its homologs from other fungi. The amino acid sequences were analyzed using MEGA 6 with neighbor-joining analysis with 1000 bootstrap replicates. Numbers on the branches represent the percentage of replicates supporting each branch. The bar represents 20% sequence divergence. (**B**) The product of four primer pairs (5F/6R, H850F/H852R, 7F/H855F, and H856F/8R) denoting the target gene, *hph*, and the recombination of upstream and downstream, respectively. (**C**) The bands of Southern blots displaying the HindIII-digested genomic DNA of *FpRtfA* deletion mutant hybridized with probe h. (**D**) The existing product of 5F/6R in RDM-C confirmed successful complementation.

We constructed the *FpRtfA* deletion mutant (RLD) using homologous recombination. The principle was to replace the entire ORF of the *FpRtfA* gene with *hph* B (Supplementary Figure S2). Following preliminary screening using hygromycin, transformants were confirmed using four primer pairs of PCR amplification. In the *FpRtfA* deletion mutant, no product was amplified using the primer pair of the ORF (*FpRtfA*-5F/6R). The other three pairs of primers used to detect the hygromycin gene, in both upstream and downstream recombination, amplified successfully (Figure 1B). In the genomic DNA of *FpRtfA* mutant strain, there was only one 4.5 kb fragment band, which had been hybridized using the *hph* probe (probe h). Therefore, there was a single locus homologous recombination in the *FpRtfA* deletion mutant (Figure 1C). The complementary strain of the reintroduction of *FpRtfA* was verified by the detected primer pair of the ORF (*FpRtfA*-5F/6R, RDM-C, Figure 1D).

### 3.2. FpRtfA Is Necessary for Normal Growth and Conidiation

To evaluate the role of *FpRtfA* in the growth and asexual development of *F. pseudo-graminearum*, the wild-type and *FpRtfA* deletion mutant strains were cultivated on a PDA medium. The colony growth of the *FpRtfA* deletion mutant strain demonstrated a slight but statistically significant reduction compared to the wild type (Figure 2A,B). In addition, the effect of *FpRtfA* on asexual development was assayed by detecting the conidial concentrations of different strains in the induced media. The number of conidia was $1.1 \times 10^7$ per mL in the wild-type strain four days post-inoculation. The corresponding amount in the deletion mutants was $3.7 \times 10^6$ per mL. This result showed that *F. pseudograminearum FpRtfA* plays an important role in conidiation (Figure 2C). Both growth and the conidiation phenomenon could be reversed as a prerequisite for reintroducing the *FpRtfA* gene into the deletion mutant strains.

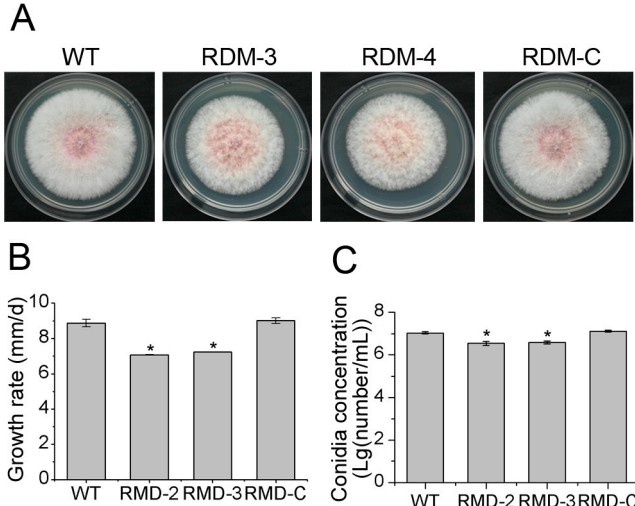

**Figure 2.** *FpRtfA* affects the growth and conidiation of *F pseudograminearum*. (**A**) Colony morphologies of the wild-type (WT), two deletion mutants (RDM-2 and RDM-3), and complementary (RDM-C) strains growing on PDA for 3 days. (**B**) Growth rates of different strains were determined by difference of radial growth between 2 and 3 days post-inoculation (dpi, growth radius per day). (**C**) The logarithm of the conidia number per milliliter after 4 dpi in induced medium. Data from three biological replicates were used to calculate the mean and standard deviation. Asterisks represent a significant difference compared to wild type ($p < 0.05$).

### 3.3. FpRtfA Affects Responses of F. pseudograminearum to Different Stress Conditions

To test whether the *FpRtfA* gene is involved in abiotic stress responses, we measured the growth rates of the wild type and deletion mutant strains on CM supplemented with NaCl (osmotic pressure), $H_2O_2$ (oxidative stress), SDS (cell membrane damaging agent), or CR (cell wall inhibitor, Figure 3A), respectively. The inhibition rates of NaCl to FpRtfA deletion mutants were increased in the presence of 0.5 M NaCl in the medium. This indicated a significant increase in the sensitivity of FpRtfA deletion mutants compared to the wild type. On the contrary, the sensitivity of *FpRtfA* deletion mutants was significantly attenuated in the presence of 3 mM $H_2O_2$ in the medium. In the presence of SDS and CR, there were no significant changes in the inhibition rates between mutants and the wild-type strain. When reintroducing the *FpRtfA* gene into the deletion mutant strain, the phenomena were restored to those of the wild type (Figure 3). These results indicated that *FpRtfA* is involved in the response to metal ions and oxidative stress in *F. pseudograminearum*.

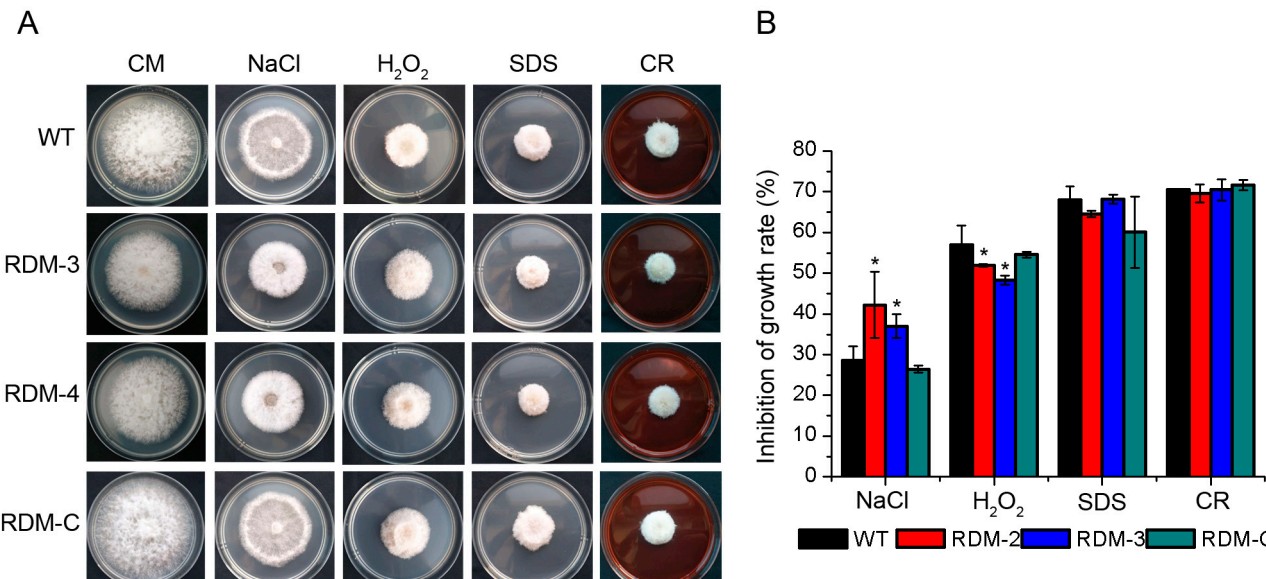

**Figure 3.** *FpRtfA* is required for responses to different stresses. (**A**) Cultures of the wild-type (WT), two deletion mutants (RDM-2 and RDM-3), and complementary (RDM-C) strains grown on MS supplemented with NaCl (Na$^+$), H$_2$O$_2$, SDS, or Congo red (CR). Images were taken after 3 dpi on different stress media. (**B**) Inhibition rate of different stresses compared with the MS without inhibitor. Bars indicate standard deviations of the mean of three replicates. Asterisks represent a statistically significant difference in inhibition compared to wild type ($p < 0.05$).

### 3.4. FpRtfA Is Required for Full Virulence in F. pseudograminearum

The virulence assays of *FpRtfA* deletion mutants and the wild type were measured using the stem base and flowering wheat heads. The wild type developed typical crown rot and head blight symptoms under the condition of artificial inoculation. However, the reduced disease index was discovered when inactivating the *FpRtfA* gene (Figure 4A,B). Observation of the coleopteres at the stem base of the wheat seedling stage showed that the coleopteres of the wild-type inoculation were full of mycelium, and most of the plant cells were decomposed and broken, while only a few mycelium could be seen in the coleopteres of the mutant, and the cells remained intact (Figure 4C). To confirm our findings, a complementation test was also undertaken with the *FpRtfA* gene. The results of this study showed that when *FpRtfA* was reintroduced into the mutant strain, the rescued phenotype was observed. These results clearly indicated that *FpRtfA* plays an important role in the virulence of *F. pseudograminearum*.

### 3.5. FpRtfA Regulates Genes Expression of Secondary Metabolites but Not DON

In order to clarify the regulatory role of the metabolic pathway affected by the *FpRtfA* gene, transcriptomes (RNA-seq) from the wild-type (raw sequence data for RNA-seq data are available in the NCBI Sequence Read Archive (SRA), accession numbers: PRJNA914495) and the *FpRtfA* deletion mutant (data are available in the NCBI SRA, accession numbers: PRJNA982303) were assessed at the genome-wide scale. In the *F. pseudograminearum* genome, there are 4047 genes that showed a significant change in expression levels in the absence of the *FpRtfA* gene. When the expression analysis at $p < 0.05$, log2Foldchange > 1 or <−1, the total number of differentially downregulated genes was 1440, while the expression of 2607 genes increased (Supplementary Figure S2). We used the Gene Ontology (GO) comprehensive database to describe either upregulated or downregulated differentially expressed gene (DEG) functions as falling into three main categories: "molecular function", "cellular components", and "biological process". Based on the GO annotations, with the threshold of significant enrichment of padj < 0.05, the majority of both upregulated and

downregulated DEGs were significantly associated with "molecular function". Cofactor binding and oxidoreductase activity were the top two terms of enriched genes. The ratios of upregulated genes to downregulated genes in these two biological processes were 150:55 and 62:19, respectively (Figure 5A). In Kyoto Encyclopedia of Genes and Genomes (KEGG) pathway analysis, the most enriched upregulated and downregulated DEGs were annotated as encoding biosynthesis of SM (Figure 5B). The GeneRatio accounted for 130 of the total 438 (Figure 5C). To verify the accuracy of the transcriptomes, the transcriptional level of ten genes encoding SM were validated using qRT-PCR. The results showed that the expression levels of five downregulated and five upregulated genes were basically the same between transcriptomes and qRT-PCR (Figure 5D).

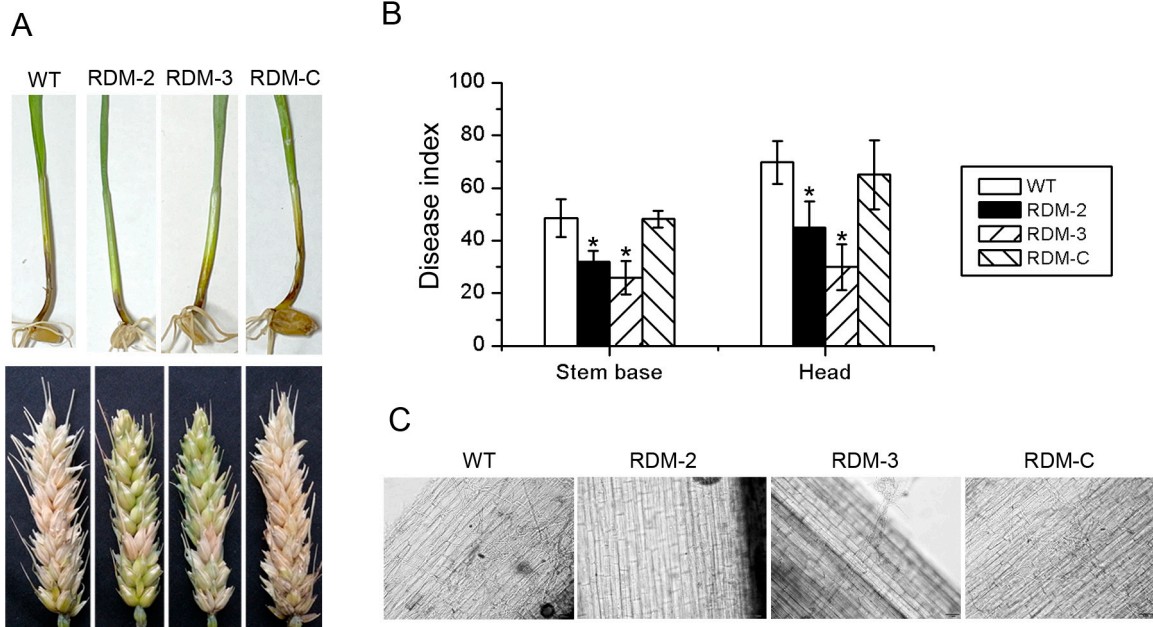

**Figure 4.** *FpRtfA* is involved in virulence against the stem base and head of wheat. (**A**) The different strains, including wild-type (WT), two deletion mutants (RDM-2 and RDM-3), and complementary (RDM-C) strains, were inoculated into the wheat stem base and flowering wheat heads. Images were taken at 21 and 20 dpi, respectively. (**B**) The disease index of the wheat stem base and heads were investigated at 21 dpi and 20 dpi, respectively. The experiments were repeated three times. Asterisks represent a significant difference comparing to the wild type ($p < 0.05$). (**C**) The infection coleoptile micrograph of different strains at the base of the wheat stem. Images were taken at 21 dpi. Bar = 1 mm.

In view of the important role of DON in the virulence of *F. pseudograminearum*, we monitored the transcription level of the TRI5 gene and production of DON in the *FpRtfA* deletion mutant and wild-type strain in inducing medium. However, the expression levels of both the TRI5 gene and DON concentration of culture solution were not significantly different compared with the wild type (Figure S3).

*3.6. A Cluster of PKS Regulated by FpRtfA Is Associated with Virulence in F. pseudograminearum*

In the differential gene expression between the trfa mutant and wild type, we found downregulated expression of one gene encoding PKS synthetase and downregulated expression of several genes in its gene cluster. We analyzed the expression pattern of all genes in this gene cluster during infection. We found that most of the genes in the mutant gene cluster were downregulated in infection (Figure 6A). We produced knock-out mutants of the key synthetize gene in this cluster (Figure 6B–D). The phenotypic results showed that the knockout of the synthetize gene did not affect the growth of the pathogen, but the

pathogenicity was weakened (Figure 6E,F). Therefore, this indicated that the regulation of the pathogenicity of TRFA might be related to this gene cluster.

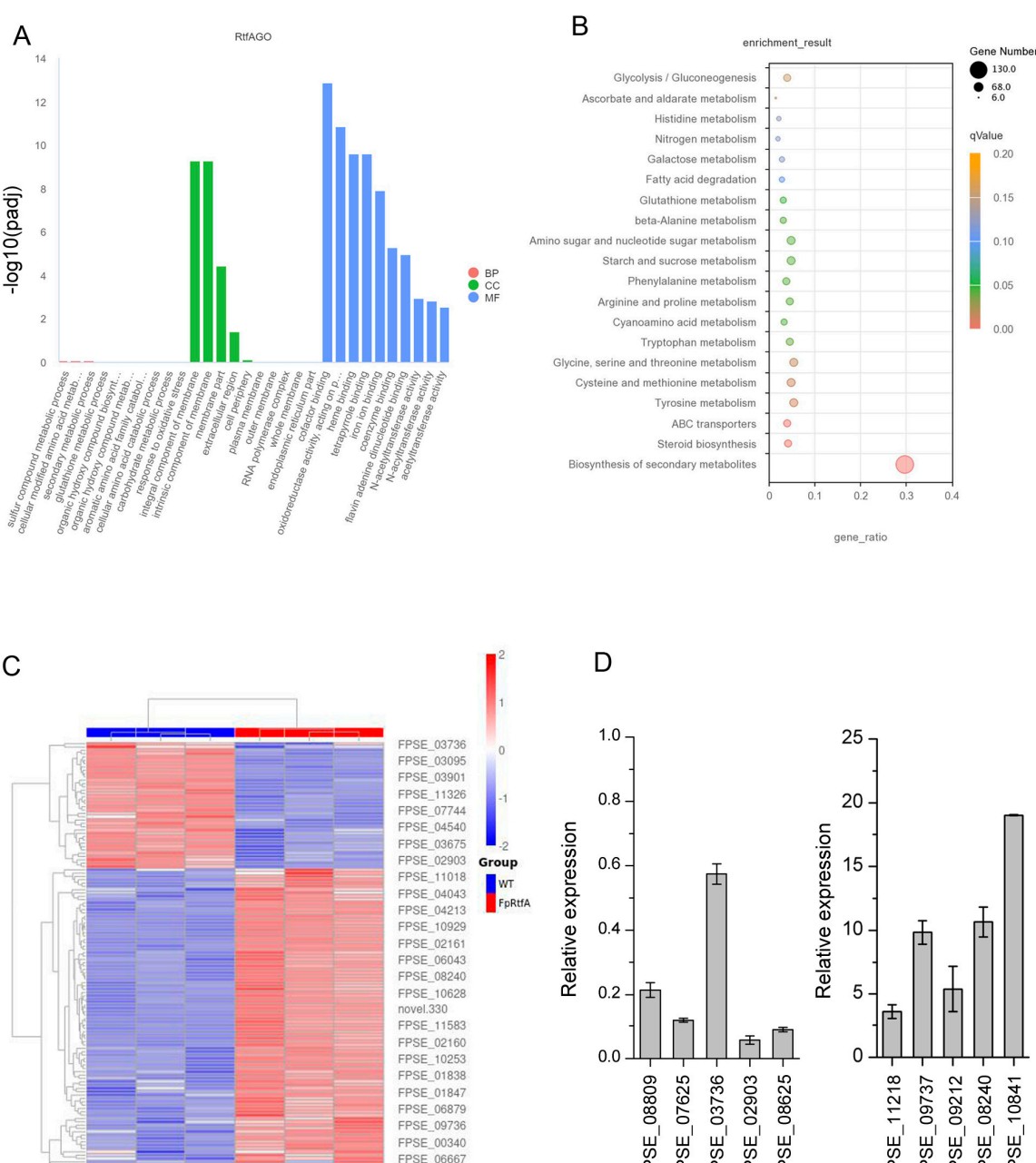

**Figure 5.** Differential expression profiles between *FpRtfA* deletion mutants and wild type. (**A**) The DEGs were categorized by their GO annotations and classified into three main categories: biological process (BP), cellular component (CC), and molecular function (MF). The differentially expressed genes (DEGs) were log2 (fold change, FC) greater than 1.0, with a threshold at a p-value and corrected *p*-value < 0.05. (**B**) Kyoto Encyclopedia of Genes and Genomes (KEGG) pathway annotation for DEGs. (**C**) Heatmap for differentially regulated genes encoding SMs. (**D**) Real-time qRT-PCR for ten differentially regulated genes encoding SM, including five upregulated genes and four downregulated genes between the wild type and *FpRtfA* mutant. The expression level of the TUB gene was used to normalize different samples. Transcript levels of wild type were arbitrarily marked as 1. The mean and standard deviation were calculated with data from three independent biological replicates. Data from three replicates were analyzed with the protected Fisher's least significant difference (LSD) test.

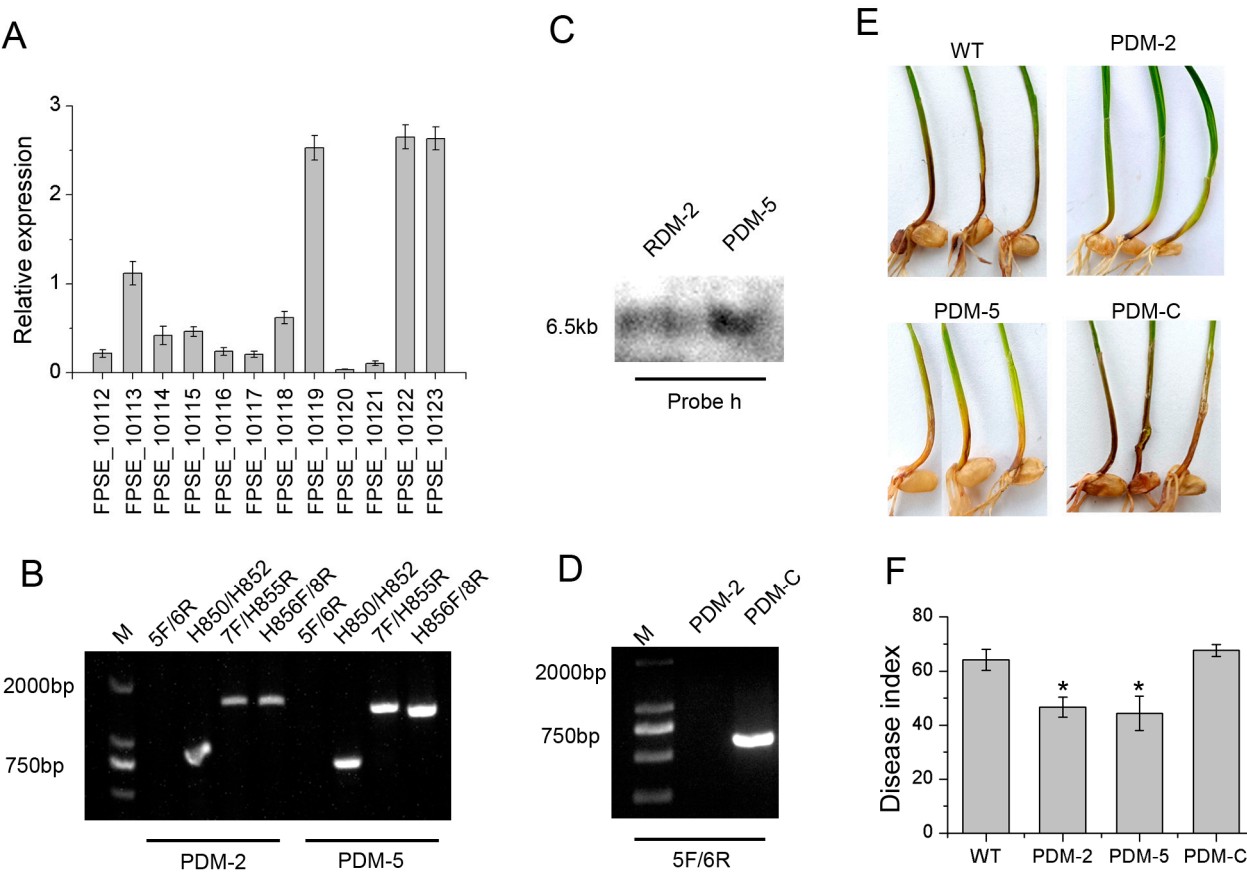

**Figure 6.** Generation of gene deletion mutants and the virulence test of the *PKS* gene in wheat. (**A**) Real-time qRT-PCR for genes encoding the SM gene cluster in which PKS is located during infection of the stem base of wheat. The sample collection date was 3 dpi. Transcript levels of conidia inoculator were arbitrarily marked as 1. The mean and standard deviation were calculated using data from three independent biological replicates. (**B**) The product of four primer pairs (5F/6R, H850F/H852R, 7F/H855F, and H856F/8R) denoting the target gene, *hph*, and the recombination of the upstream and downstream, respectively. (**C**) The bands of the Southern blots showed that the HindIII-digested genomic DNA of the *FpPKS* deletion mutant hybridized with probe h. (**D**) The existing product of 5F/6R in PDM-C confirmed successful complementation. (**E**) The different strains, including wild type (WT), two deletion mutants (PDM-2 and PDM-3), and complementary (PDM-C), were inoculated into the wheat stem base. Images were taken at 21 dpi. (**F**) The disease index of the wheat stem base was investigated at 21 dpi. The experiments were repeated three times. Asterisks represent a significant difference compared to the wild type ($p < 0.05$).

## 4. Discussion

*Fusarium pseudograminearum* is an agriculturally relevant pathogen. Adverse effects such as yield loss and toxin contamination can be caused by this pathogen, and it is crucial to identify potential regulators that can be utilized in innovative strategies aimed at reducing the survival, dissemination, production of toxic compounds, and virulence of *F. pseudograminearum* [41,42]. Among these regulators, a putative RNA-Pol II transcription elongation factor protein, RtfA, has been identified as being required for ubiquitination of histone H2B in *Saccharomyces cerevisiae* [18]. RtfA has also been shown to regulate the production of several secondary metabolites (SMs) in *Aspergillus* spp. [19–21]. To investigate its regulatory role in the development and secondary metabolites (SMs) of *F. pseudograminearum*, we deleted the RtfA homologue, *FpRtfA*, in this soil-borne pathogen. The present study revealed that the deletion of *FpRtfA* has a mild impact on the mycelial growth rate in *F. pseudograminearum*. This effect was similar to that in *A. nidulans* and

*A. flavus*; the decrease in the growth rate was minor [19,21]. However, a more pronounced effect on the growth of the *RtfA* deletion mutant was seen in *A. fumigatus* [20]. In addition, the effect on conidiation of the absence of RtfA was different in the three types of *Aspergillus* spp. In *A. nidulans* and *A. flavus*, a lack of *RtfA* resulted in impaired conidiation. In our study, the effect of FpRtfA on conidiation was a slight but statistically significant reduction. A different effect was observed in *A. fumigatus*, where the *RtfA* deletion mutant strain exhibited a significant increase in conidial production compared to the wild type [20–22]. These findings suggested that the function of RtfA may be species-specific in fungi, and its roles could diverge among different fungal species.

In the course of plant infection, pathogens are exposed to various environmental and host stresses, such as metal ions, cell wall and membrane stress, and oxidative stress [43]. For this reason, we conducted stress tests of *F. pseudograminearum* deletion mutants and wild-type strains with various inhibitors. Our results showed that the deletion of *FpRtfA* affected sensitivity to NaCl and $H_2O_2$ instead of SDS and CR in different forms. When *FpRtfA* was knocked out, *F. pseudograminearum* became more susceptible to NaCl stress, while exhibiting increased resistance to oxidative stress caused by $H_2O_2$. In the transcriptome analysis of this study, the differentially expressed genes (DEGs) were enriched into response to oxidoreductase activity and iron ion binding based on GO enrichment statistics may be related to the stress response of $H_2O_2$ and NaCl, respectively. Our results were different from the effect of RtfA on *A. fumigatus*. The suppressed growth of the *RtfA* mutant on 20 µM of menadione medium showed that the deletion of *RtfA* results in greater sensitivity to oxidative stress in *A. fumigatus* [20]. However, we have not yet demonstrated the relationship between sensitivity of abiotic stress and virulence in *F. pseudograminearum*.

The present investigation also demonstrated the role of the *FpRtfA* gene for normal virulence in *F. pseudograminearum*. An identical regulatory effect was demonstrated in *A. flavus*. A plant and an animal model were used to demonstrate that RtfA is relevant to pathogenesis in *A. flavus*. The deletion strain showed a reduction in conidiation when infecting peanut seeds compared to the wild-type strain. A reduction in virulence was also observed in an animal infection when *RtfA* was deleted [22]. Similarly, RtfA is relevant in *A. fumigatus* animal infections [20]. More importantly, the modes of action that contributed to the observed decrease in virulence were expounded in *A. flavus*. The factors included delay and reduction in adherence, regulation of hydrolytic activity, cell wall composition and oxidative stress resistance, and production of potent mycotoxins important for normal fungal invasion and colonization of the host [22]. In the present study, the reason for the reduction of virulence was resolved using genome-wide expression analysis. A quarter of the genes were affected in the *F. pseudograminearum* genome when *FpRtfA* was deleted. This regulation explained multiple impacts of the phenotype in the *FpRtfA* deletion mutant. Based on GO enrichment statistics, DEGs were categorized into carbohydrate metabolic processes (Supplementary Table S3). The carbohydrate metabolic process also leads to growth reduction in *Penicillium expansum* [44]. Therefore, the effect on the growth rate of *FpRtfA* deletion may be due to the regulation of the carbohydrate metabolic process in *F. pseudograminearum*. Similarly, multiple DEGs can be categorized into two biological processes and molecular functions related to the response to oxidative stress and oxidoreductase activity, acting on paired donors, with incorporation or reduction of molecular oxygen. This finding seemed consistent with the reduced sensitivity to $H_2O_2$ of the mutant. However, all these results are predictions, and the functions of many genes need further testing.

The regulation of secondary metabolites (SMs) by *RtfA* has been reported in several *Aspergillus* spp. Studies. In *A. nidulans*, the production of sterigmatocystin and penicillin biosynthesis was positively regulated by *RtfA*, while the production of tryptoquivaline F, pseurotin A, fumiquinazoline C, festuclavine, and fumigaclavines A, B, and C was negatively regulated by *RtfA* in *A. fumigatus*. Moreover, *RtfA* regulates the synthesis of AFB1 in *A. flavus*, and the synthesis of aflatrem, paxilline, leporins, aflavinines, ditryptophenaline, and aspirochlorine is also controlled by RtfA in this fungus [20–22]. In the present study,

we demonstrated that *FpRtfA* regulates SMs. The highest expression levels and the most DEGs were in the biosynthesis of the SM pathway in KEGG pathway analysis. We also investigated the impact of *FpRtfA* on DON, a significant virulence factor in *F. pseudogramin-earum* infection [12–14]. However, *FpRtfA* was dispensable for DON generation, showing that the regulation of virulence by *FpRtfA* was unrelated to DON. Fortunately, we found one downregulated expression gene encoding PKS synthetase in an SM gene cluster of *FpRtfA* deletion mutant involved virulence. Hence, it is plausible to infer that FpRtfA exerts its regulatory effect on the virulence of pathogens by modulating the expression of this PKS gene. However, the pathways in which FpRtfA regulated secondary metabolism need to be further studied.

## 5. Conclusions

In this study, we identified that FpRtfA is a regulator of vegetative growth, conidiation, sensitivity to NaCl and $H_2O_2$, and virulence in *F. pseudograminearum*. More importantly, many downstream genes related to the biological processes described above were detected using genome-wide gene expression analysis. The positive regulation of SMs, including PKS, could be related to virulence. Future research should focus on the regulation model of FpRtfA for SMs.

**Supplementary Materials:** The following supporting information can be downloaded at: https://www.mdpi.com/article/10.3390/su151411401/s1, Figure S1: Structures of *FpRtfA*/FpRtfA and homology comparison between *F pseudograminearum* and *A. nidulans*. (A) The *FpRtfA* gene consists of 2031-bp, interrupted by three intron, and encodes a predicted protein of 597 amino acids with a Plus-3_dom domains. (B) Phylogenetic analysis of FpRtf of *V. mali* and its homologs from other fungi. The amino acid sequences were analyzed using MEGA 6 with neighbor-joining analysis with 1000 bootstrap replicates. Numbers on the branches represent the percentage of replicates supporting each branch. The bar represents 20% sequence divergence. Figure S2: Gene deletion schematic of *FpRtfA*. The double-joint method was used to generate the replacement of *FpRtfA* gene. The small arrows sign the directions and positions of primers used for amplifying recombinant fragment and detecting target gene. Figure S3: Transcription level of TRI5 gene and DON concentration in inducing medium. (A) Relative transcript abundances of TRI5 gene in mycelium at inducing medium were compared between the wild-type and *FpRtfA* mutants at 7 dpi. Expression level of TUB gene was used to normalize different samples. Transcript levels of wild type were arbitrary assigned to 1. (B) DON concentration of culture solution at inducing medium. The mean and standard deviation were calculated with data from three independent biological replicates. Table S1: Primers used for deletion, complementation and gene expression. Table S2: All differentially expressed genes between FpRtfA deletion mutant and wild type. Table S3: The main categories of differentially expressed genes on the GO annotations. Table S4: The enriched pathway of differentially expressed genes on the KEGG annotations.

**Author Contributions:** Conceptualization, Y.W. (Yuxing Wu) and L.K.; methodology, Y.W. (Yuxing Wu), Y.W. (Yajiao Wang), S.H., and Q.L.; validation, Y.W. (Yajiao Wang) and S.H.; investigation, Y.W. (Yuxing Wu) and S.H.; data curation, Y.W. (Yuxing Wu) and Q.L.; writing—original draft preparation, Y.W. (Yuxing Wu); writing—review and editing, L.K.; supervision, L.K.; project administration, Y.W. (Yuxing Wu) and L.K.; funding acquisition, L.K. All authors have read and agreed to the published version of the manuscript.

**Funding:** This research was funded by the Basic Research Funds of Hebei Academy of Agriculture and Forestry Sciences, grant number 2021120201; National Key R&D Program of China, grant number 2022YFD1901300; Natural Science Foundation of Hebei, grant number C2021301042; and the HAAFS Agriculture Science and Technology Innovation Project, grant number 2022KJCXZX-ZBS-7.

**Institutional Review Board Statement:** Not applicable.

**Informed Consent Statement:** Not applicable.

**Data Availability Statement:** The original contributions presented in the study are included in the article/Supplementary Materials. Further inquiries can be directed to the corresponding author.



**Conflicts of Interest:** The authors declare no conflict of interest.

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
