# Peer review of "RNA-Pol II Transcription Elongation Factor FpRtfA Is Required for Virulence and Secondary Metabolism in Fusarium pseudograminearum"

_sustainability, doi:10.3390/su151411401_

Round 1
Reviewer 1 Report
the study concerns an important species of wheat in many countries of the world and the risk of spreading selected genotypes of the fungus Fusiarum. The results of the research indicate the development of individual genotypes and their ability to infect plants, the strength of the infection. This is valuable and important information for cereal producers and plant protection workers. It can be helpful in creating new, highly specialized plant protection products.
Reviewer 2 Report
Manuscript Title : RNA-Pol II transcription elongation factor FpRtfA is required for growth, development, secondary metabolism, and virulence in Fusarium pseudograminearum
Manuscript ID : Sustainability-2476081
Authors :Yuxing Wu , Yajiao Wang , Sen Han , Qiusheng Li , Lingxiao Kong *
Recommendation : Suitable for publication with minor revision
Comments:
The present article, “RNA-Pol II transcription elongation factor FpRtfA is required for growth, development, secondary metabolism, and virulence in Fusarium pseudograminearum”, involves RNA-Pol II transcription elongation factor, FpRtfA for regulating the secondary metabolites (SMs) in Aspergillus spp. FpRtfA has important 20 roles in growth, asexual development, and the response to metal ions and oxidative stress. Overall, the organization of the manuscript is good but still minor change are required to publish in its present form in Sustainability. The specific comments is given below.
1. FpRtfA affects the responses to different stress conditions. The author needs to provide the possible explanation for all of the stress condition. Why does NaCl and H2O2 showing sensitive instead of SDS and CR?
Reviewer 3 Report
Thank the authors for this interesting paper. I send some recommendations according to my opinion it can improve the article:
- the title is very difficult
- in abstract I miss the main aim, the purpose of the article
- the theoretical background should be extended, more studies in this topic are needed because this is a very difficult topic
- the methods are great and the results as well, everything is described in a detail
- the section discussion compares the findings with the another authors, but I miss the limitations
- Conclusions are needed
